# NuMI Beam Monitoring Simulation and Data Analysis †

**Yiding Yu** [1,*] **, Thomas Joseph Carroll** [2] **, Sudeshna Ganguly** [3] **, Karol Lang** [4] **, Eduardo Ossorio** [1] **, Pavel Snopok** [1] **, Jennifer Thomas** [5] **, Don Athula Wickremasinghe** [3] **and Katsuya Yonehara** [3]

1. Department of Physics, Illinois Institute of Technology, Chicago, IL 60616, USA
2. Department of Physics, University of Wisconsin–Madison, Madison, WI 53706, USA
3. Fermi National Accelerator Laboratory, Batavia, IL 60510, USA
4. Department of Physics, University of Texas at Austin, Austin, TX 78712, USA
5. Physics and Astronomy Department, University College London, Gower Street, London WC1E 6BT, UK
* Correspondence: yyu79@hawk.iit.edu
† Presented at the 23rd International Workshop on Neutrinos from Accelerators, Salt Lake City, UT, USA, 30–31 July 2022.

**Abstract:** Following the decommissioning of the Main Injector Neutrino Oscillation Search (MINOS) experiment, muon and hadron monitors have emerged as vital diagnostic tools for the NuMI Off-axis $\nu_\mu$ Appearance (NOvA) experiment at Fermilab. These tools are crucial for overseeing the Neutrinos at the Main Injector (NuMI) beam. This study endeavors to ensure the monitor signal quality and to correlate them with the Neutrino beam profile. Leveraging muon monitor simulations, we systematically explore the monitor responses to variations in proton-beam and lattice parameters. Through the amalgamation of individual pixel data from muon monitors, pattern-recognition algorithms, simulations, and measured data, we devise machine-learning-based models to predict muon monitor responses and Neutrino flux.

**Keywords:** NuMI beam; simulation; muon monitor; pixel study

## 1. Overview of the NuMI Beamline Monitoring System

The NOvA experiment [1] utilizes Fermilab's NuMI Neutrino beam [2]. Generated by 120-GeV protons from the Main Injector interacting with a 1.2 m long graphite target, the beam is refined with two magnetic horns that focus the resultant pions and kaons. These mesons then decay within a 675 m long pipe, predominantly into muons and muon neutrinos. The beamline's design is depicted in Figure 1. Positioned downstream of the hadron absorber are three muon monitors (MM1, MM2, MM3), each consisting of 9 × 9 pixel ionization chamber arrays. Since the decommissioning of the MINOS detector in February 2019, these muon monitors have become vital for beam oversight and alignment, ensuring optimal performance. This continuous monitoring is pivotal for the NOvA experiment's success. The setup also features a hadron monitor upstream of the absorber, several beam-position monitors, and toroids overseeing the primary proton beam.

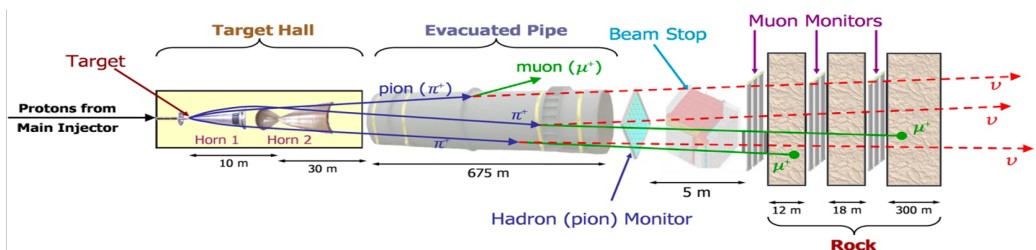

**Figure 1.** Schematic of the NuMI beamline [3] featuring three muon monitors and two magnetic horns.

## 2. Introduction to Muon Monitor Simulation

The muon monitor simulation utilizes g4NuMI [4], a simulation tool grounded in Geant4 [5]. Within G4NuMI, we model every component of the NuMI beamline, encompassing the target, magnetic horns, decay pipe, hadron monitor, and muon monitors.

Our simulation procedure for the muon monitor consists of the following steps:

1. Neutrinos are generated using G4NuMI.
2. These neutrinos are then traced back to their originating particles.
3. Muons are subsequently produced from these neutrinos and their parent entities.
4. These muons are introduced into a distinct G4NuMI environment to generate readings at the muon monitors.

Figure 2 illustrates a 2D histogram of MM1. In the simulation, each of its 81 bins equates to a pixel indicating the count of muon events. Conversely, for actual data, the pixels capture the voltage readings from the muon monitors.

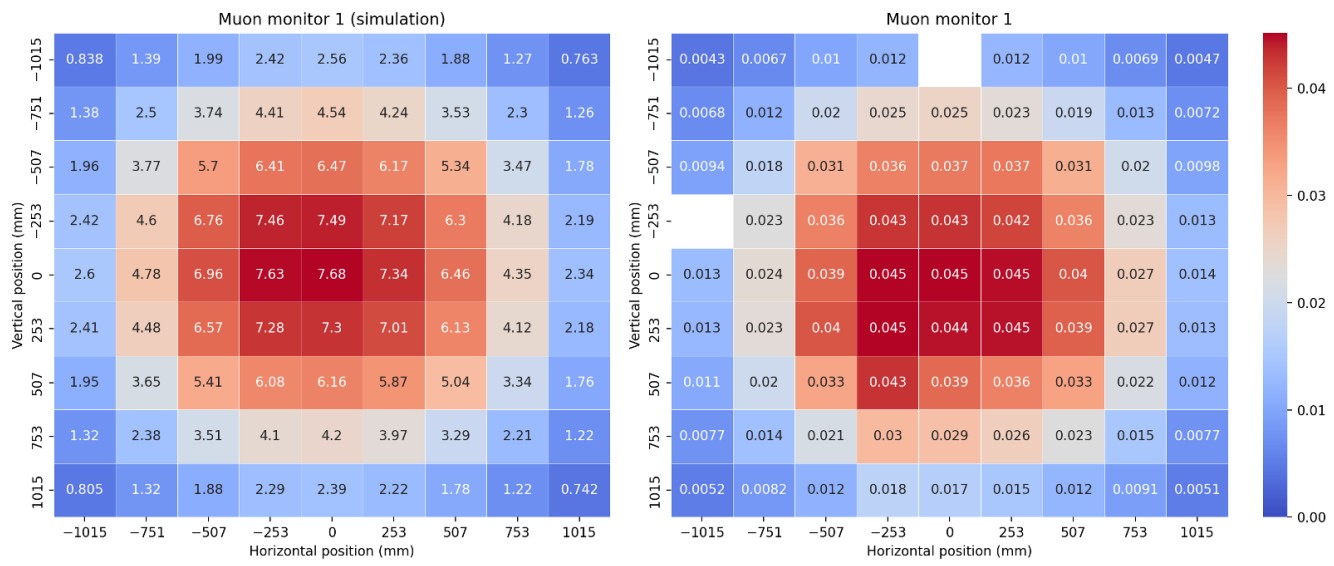

**Figure 2.** Two−dimensional histogram at MM1. On the (**left**) (simulation), pixels indicate event counts divided by $1 \times 10^5$. On the (**right**) (measurement), pixels display voltage signals normalized to beam intensity.

In our muon monitor simulation, the momentum distribution across the 81 pixels provides a detailed correlation between muon distribution and the primary proton-beam profile. Figure 3 illustrates this: spectra differ between pixels, with momentum peaks declining as one transitions from MM1's center to its edge. Each pixel's spectra in the diagram are marked, with the central row as X1 to X9 and the central column as Y1 to Y9.

As highlighted in Figure 1, MM1, MM2, and MM3 are strategically placed downstream of the NuMI hadron absorber, with intervals of 12 and 18 m of rock between them. This spacing offers sensitivity to various muon momenta. When validating the MM1 simulation against real-world measurements, we encountered computational hurdles. Our resolution was to integrate two innovative techniques: uniform beam and multiple decay.

The uniform beam method simulates a uniformly distributed proton beam striking the target. During analysis, we then apply Gaussian weights to the protons based on their positions to mimic varying beam locations.

On the other hand, the multiple decay technique compels each target-produced pion to decay into several muons. Validity checks confirmed this method's accuracy, comparing the results of up to 75× decays per pion against a reference simulation, revealing a solid agreement.

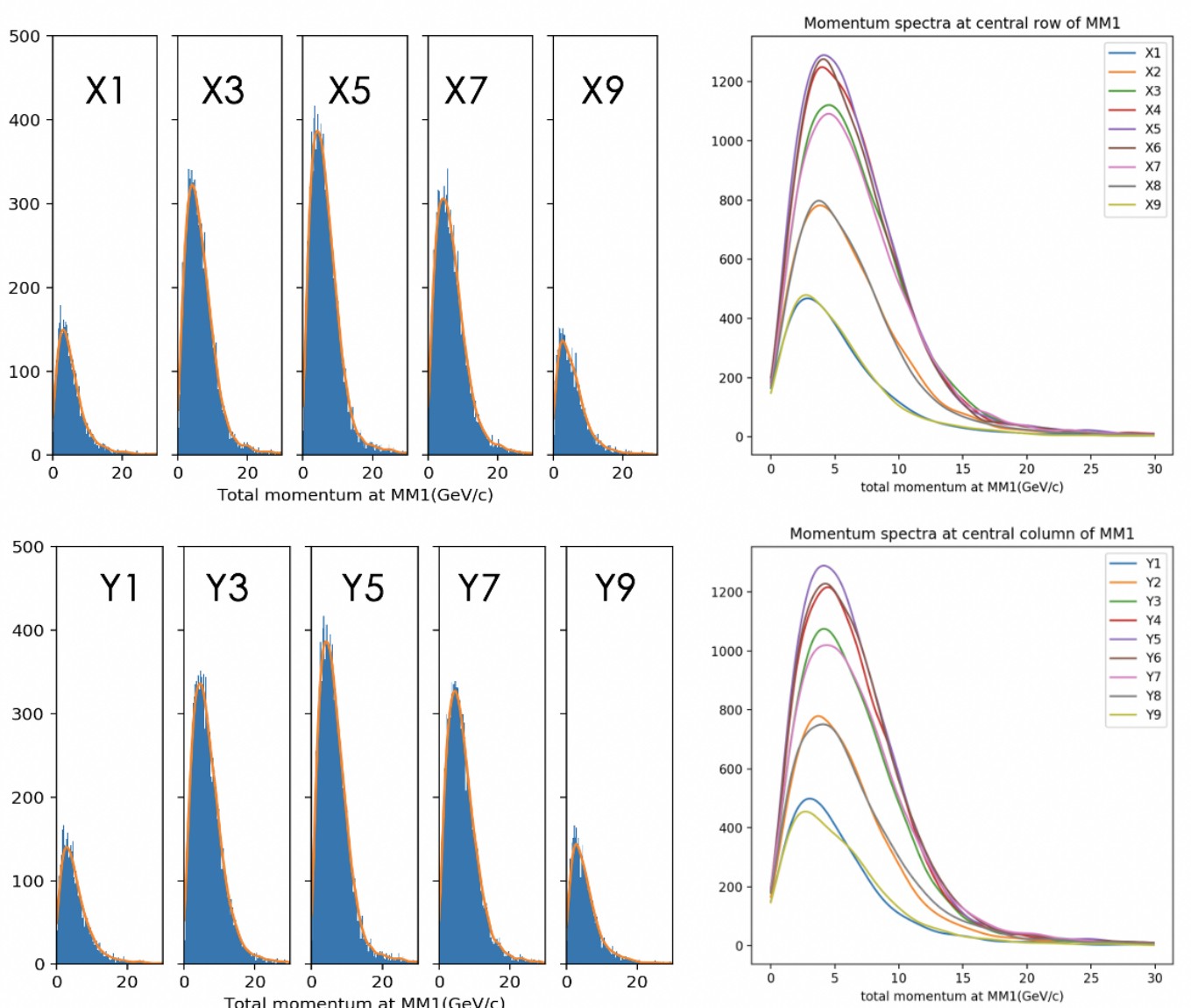

**Figure 3.** The plot illustrates the muon momentum spectra for MM1 pixels, organized in a central row (X1–X9) and column (Y1–Y9). Noticeable is the shift of peaks to lower momentum as one moves from the center to the edge. Different color lines in the right plots represent spectra for distinct pixels.

Combined, these techniques empower us to generate samples with remarkably high statistical precision.

## 3. Comparison between Simulation and Measurement Data

To understand the NuMI beam's behavior and be able to predict the effect of changes on the key beam parameters, multiple beam scans were carried out. The beam position on target, beam spot size, and focusing horn currents were changed in a controlled fashion. Beam scans indicated how each MM responded to beam-position and horn-current variations.

Our interest is the related change of muon events at muon monitors for different proton-beam positions. We compared the measurement data and the simulation results by looking at muon beam centroids and individual pixels (see Figures 4 and 5). The simulation results exhibit a consistent behavior with the measurement data. Specifically, the centroids of the muon beam on MM1 and MM3 show opposite slopes as a function of the proton-beam position on the target. And for all operational pixels at MM1, the comparison shows

that the simulation largely aligns with data trends, albeit with some quantitative disparities. Two pixels on the top and left of MM1 are broken and non-operational, leading to missing data in the comparison.

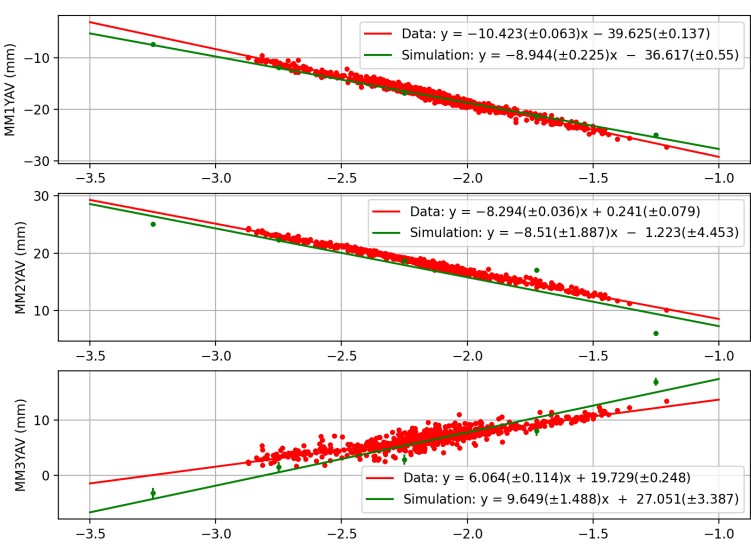

**Figure 4.** Slopes of muon beam position at MM1 to MM3 plotted against proton−beam position on the target, comparing data and simulation results.

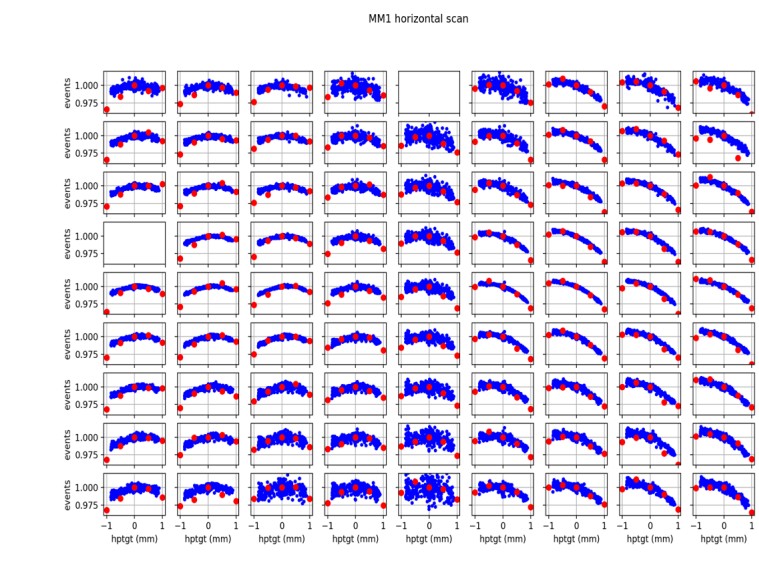

**Figure 5.** Comparison of MM1 pixels for normalized voltage signals and muon event counts across varying horizontal target beam positions. Blue represents data, while red indicates simulation.

## 4. Beam Parameter Effect on Pixels at MMs

To study the effect of beam parameters, we compared the simulation results with changed parameters for each pixel at MM1–MM3. Each simulation had 50 million protons on target (PoT) and each pion and kaon had multiple decays to muons. The comparison is illustrated using a 2D histogram, where the number in each bin represents the ratio of muon events.

### 4.1. Horn1 Tilt Angle (x Axis)

In Figure 6, we adjusted the Horn1 tilt angle (represented by the x-axis) from −3 mrads to +3 mrads, indicating Horn1's downward and upward tilting. Figure 7, featuring 2D ratio histograms, illustrates the influence of varying Horn1 tilt angles across MM1–MM3. Each pixel's value in these histograms represents the ratio of muon events from simulations with distinct Horn1 tilt settings to those from a reference simulation. The simulation outcomes highlight MM2 and MM3's heightened sensitivity to Horn1 tilt changes, with the top pixels undergoing notably more significant variations than the bottom ones.

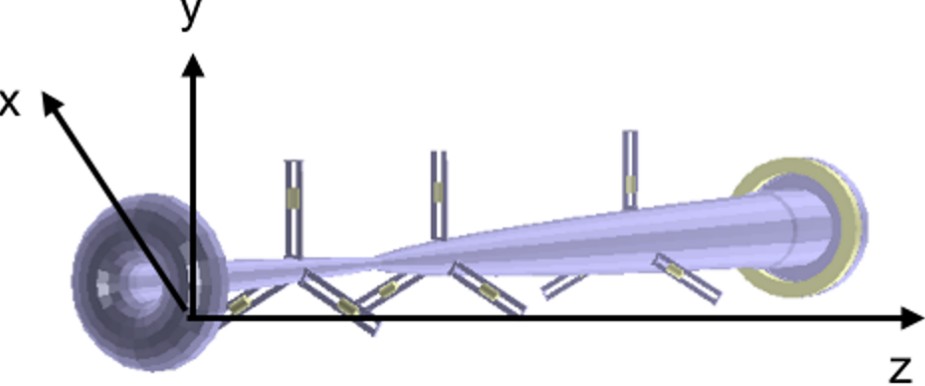

**Figure 6.** Model of Horn1.

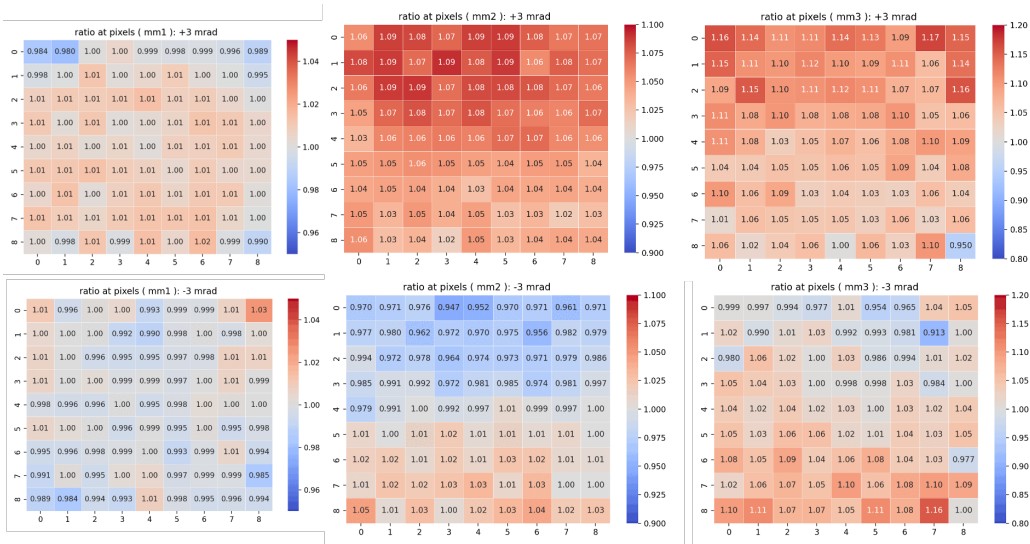

**Figure 7.** Two-dimensional histograms of ratios (muon events) at MM1–3 for tilt angle −3 mrads (**top**) and −3 mrads (**bottom**).

### 4.2. Target Offset in Vertical Direction

The target vertical offset was adjusted between −1 mm and +1 mm, revealing a linear correlation between muon events at pixels and the offset in Figure 8. Two specific pixels, 73 (center) and 70 (near the edge), were selected to analyze event variations, see Figure 9. While all three muon monitors exhibited similar patterns, MM1 was particularly sensitive to the target vertical offset.

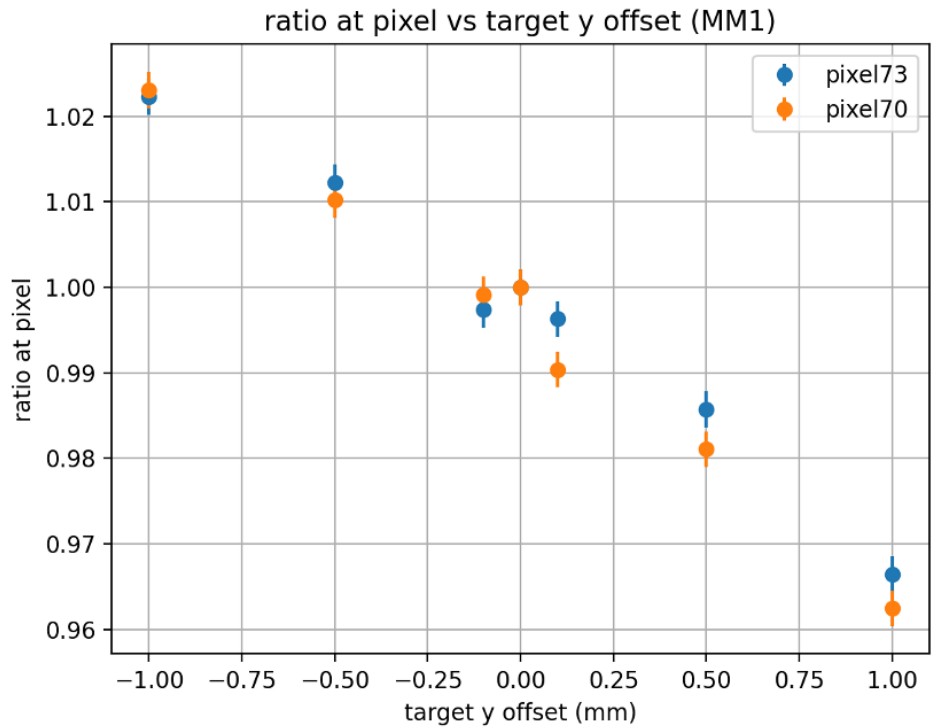

**Figure 8.** Ratios of pixels vs. target vertical offset.

| 33 | 42 | 51 | 60 | 69 | 78 | 6 | 15 | 24 |
|----|----|----|----|----|----|----|----|----|
| 34 | 43 | 52 | 61 | 70 | 79 | 7 | 16 | 25 |
| 35 | 44 | 53 | 62 | 71 | 80 | 8 | 17 | 26 |
| 36 | 45 | 54 | 63 | 72 | 0 | 9 | 18 | 27 |
| 37 | 46 | 55 | 64 | 73 | 1 | 10 | 19 | 28 |
| 38 | 47 | 56 | 65 | 74 | 2 | 11 | 20 | 29 |
| 39 | 48 | 57 | 66 | 75 | 3 | 12 | 21 | 30 |
| 40 | 49 | 58 | 67 | 76 | 4 | 13 | 22 | 31 |
| 41 | 50 | 59 | 68 | 77 | 5 | 14 | 23 | 32 |

**Figure 9.** Pixel map of muon monitors.

*4.3. Other Parameters' Effects*

For the change of different parameters, we used the improved simulation to check whether the muon monitors can catch the effect of the parameters. It indicates the muon monitors are able to find the effect of those changes in Figure 10.

| Beam parameter | change | MM1 | MM2 | MM3 |
|---|---|---|---|---|
| Horn1 tilt angle Y | ± 3 mrad | ✘ | ○ | ○ |
| Horn1 tilt angle Y | ± 2 mrad | ✘ | ○ | ○ |
| Horn1 tilt angle Y | ± 1 mrad | ✘ | ○ | ✘ |
| Target offset Y | ± 1 mm | ○ | ○ | ○ |
| Target offset Y | ± 0.5 mm | ○ | ○ | ○ |
| Target offset Y | ± 0.1 mm | ○ | ✘ | ✘ |
| Target density | −1% | ✘ | ✘ | ✘ |
| Horn1 position Y | ± 1 mm | ✘ | ○ | ○ |
| Horn current | ± 1 kA | ○ | ○ | ○ |
| Magnetic field in decay pipe | No and double | ○ | ○ | ✘ |

**Figure 10.** Circles indicate differences exceeding 3 sigma at certain pixels, while crosses represent differences within 3 sigma. Here, sigma is the difference in the simulation divided by the statistical error of the simulation.

## 5. Machine Learning Algorithms Based on Muon Monitor Data

In addition to analyzing MM data and simulation results, we built a machine learning (ML) model that predicts/reconstructs primary beam parameters and the magnetic horn current by using muon monitor data. The model is trained using randomly selected past data samples. ML predictions can be used to monitor beamline issues in the future. The plan is to implement the ML model predictions for daily NuMI beamline data monitoring and catching common failure modes.

**Author Contributions:** Conceptualization, Y.Y., T.J.C., K.L., P.S., J.T., D.A.W. and K.Y.; methodology, Y.Y., T.J.C., K.L., P.S., J.T., D.A.W. and K.Y.; software, Y.Y. and T.J.C.; validation, Y.Y. and E.O.; investigation, Y.Y.; resources, D.A.W. and K.Y.; data curation, E.O.; writing—original draft preparation, Y.Y.; writing—review and editing, Y.Y., T.J.C., S.G., J.T. and K.Y.; visualization, Y.Y.; supervision, K.L., P.S., J.T. and K.Y.; project administration, P.S. All authors have read and agreed to the published version of the manuscript.

**Funding:** This work is supported by the Fermi Research Alliance, LLC manages and operates the Fermi National Accelerator Laboratory pursuant to Contract number DE-AC02-07CH11359 with the United States Department of Energy.

**Institutional Review Board Statement:** FERMILAB-CONF-23-438-STUDENT-V.

**Informed Consent Statement:** Not applicable.

**Data Availability Statement:** Data is unavailable due to privacy.

**Acknowledgments:** This manuscript has been authored by Fermi Research Alliance, LLC under Contract No. DE-AC02-07CH11359 with the U.S. Department of Energy, Office of Science, Office of High Energy Physics.

**Conflicts of Interest:** The authors declare no conflict of interest.

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
