# Peer review of "NuMI Beam Monitoring Simulation and Data Analysis†"

_psf, doi:10.3390/psf8010073_

Round 1
Reviewer 1 Report
Comments and Suggestions for Authors
Please address all the comments below.
Figure 1, please add a reference. It was originally from Zarko Pavlovic. Please highlight the number of the MMs and which is Horn 1, all of which will be mentioned later in the content.
Lines 28-29 should be removed.
Line 35, please elaborate on what is being compared here.
Figure 2, please remake the left figure, so it does not look like an incomplete screenshot. For the right figure, please reduce the number of digits in the plot, also, its title is covered by the figure. Please provide more details in the title to explain both figures.
Figure 3, please define X1-9 and Y1-9. The right figures are too small to see. Again, please provide more details in the title to explain all figures.
Line 38, add a space in front of "One".
Lines 39, 63, 73, 78, please fix the "??"!
Line 44, where do we draw the conclusion about the good agreement?
Line 55, stat -> statistics.
Line 64, add "." after "data".
Figure 4, please define what variables are being shown here. Please move the legends so they do not cover the data points. Again, please provide more details in the title to explain both figures.
Figure 5, please define what variables are being shown here. What does "latest" in the title mean? Please explain the empty spots.
Line 68, "MM" is suddenly replaced by "muon monitor" in Sec. 4. Does "1 3" mean 1 and 3? Is "imulation" a typo? Define "MPoT".
Line 69, please fix the typo.
Line 70, please fix the typo.
Line 71, please define the x-axis first.
Figure 6, please change "mm" to "MM" in all the titles. Again, please provide more details in the title to explain all figures.
Line 77, shows -> show.
Figure 8, why pick these 2 specific pixels?
Figure 10, "3 sigma" from what?
Line 84, add ~ after "Fig.".
Author Response
Dear Reviewer,
Thank you for your thoughtful comments and suggestions regarding my manuscript. I have taken them into consideration and made the necessary revisions to address each point you raised.
Specifically, I've corrected the typographical errors and regenerated the relevant plots to ensure clarity and accuracy. I've also added the necessary details as you recommended to provide a more comprehensive understanding of the plots.
I apologize for the delay in my response. The past two weeks have been particularly demanding as I've been deeply engrossed in finalizing my thesis and preparing for my thesis defense.
Please find the revised version of the manuscript attached for your review.
